# Urine lipoarabinomannan concentrations among HIV-negative adults with pulmonary or extrapulmonary tuberculosis disease in Vietnam

**Nguyen B. Hoa**[1⊙], **Mark Fajans**[2⊙], **Hung Nguyen Van**[1], **Bao Vu Ngoc**[3], **Nhung Nguyen Viet**[1], **Hoa Nguyen Thi**[1], **Lien Tran Thi Huong**[3], **Dung Tran Minh**[3], **Cuong Nguyen Kim**[1], **Trinh Ha Thi Tuyet**[1], **Tri Nguyen Huu**[1], **Diep Bui Ngoc**[4], **Hai Nguyen Viet**[1], **An Tran Khanh**[3], **Lorraine Lillis**[5], **Marcos Perez**[5], **Katherine K. Thomas**[6], **Roger B. Peck**[5], **Jason L. Cantera**[7], **Eileen Murphy**[5], **Olivia R. Halas**[5], **Helen L. Storey**[5], **Abraham Pinter**[8], **Morten Ruhwald**[9], **Paul K. Drain**[2,6,10], **David S. Boyle**[5]\*

1 National Lung Hospital, Hanoi, Vietnam, 2 Department of Epidemiology, University of Washington, Seattle, Washington, United States of America, 3 Southeast Asia Hub, PATH, Hanoi, Vietnam, 4 Center for Creative Initiatives in Health and Population, Hanoi, Vietnam, 6 Department of Global Health, University of Washington, Seattle, Washington, United States of America, 5 Diagnostics Global Program, PATH, Seattle, Washington, United States of America, 7 Global Health Labs, Bellevue, Washington, United States of America, 8 Public Health Research Institute Center, New Jersey Medical School, Rutgers University, New Brunswick, New Jersey, United States of America, 9 TB Program, Foundation for Innovative New Diagnostics, Geneva, Switzerland, 10 Department of Medicine, University of Washington, Seattle, Washington, United States of America

⊙ These authors contributed equally to this work.
\* dboyle@path.org

**Data Availability Statement:** All of the data generated in this study can be publicly accessed at

## Abstract

Lipoarabinomannan (LAM) is a promising target biomarker for diagnosing subclinical and clinical tuberculosis (TB). Urine LAM (uLAM) testing using rapid diagnostic tests (RDTs) has been approved for people living with HIV (PLWH), however there is limited data regarding uLAM levels in HIV-negative (HIV-ve) adults with clinical TB. We conducted a clinical study of adults presenting with clinical TB-related symptoms at the National Lung Hospital in Hanoi, Vietnam. The uLAM concentrations were measured using electrochemiluminescent (ECL) immunoassays and compared to a microbiological reference standard (MRS) using GeneXpert Ultra and TB culture testing. Estimated uLAM concentrations above plate specific calculated limit of detection (LOD) were considered uLAM positive. Additional microbiological testing was conducted for possible extrapulmonary TB (EPTB). Among 745 participants enrolled, 335 (44.9%) participants with presumptive pulmonary TB (PTB) and 6 (11.3%) participants with presumptive EPTB had confirmed TB disease. Overall, the S/A antibody pair had a sensitivity of 39% (95% Confidence Interval [CI] 0.33, 0.44) and a specificity of 97% (95% CI 0.96, 0.99) compared to the MRS. The F/A antibody pair had a sensitivity of 41% (95% CI 0.35, 0.47) and a specificity of 79% (95% CI 0.75, 0.84). S/A provided greater discriminatory ability compared to F/A for both individuals with presumptive PTB (AUROC: 0.74 vs 0.63, $p<0.0001$) and presumptive EPTB (0.76 vs 0.54, $p = 0.045$) when using the MRS. Among HIV-ve participants in an adult cohort in Vietnam, the concentrations

Dataverse https://doi.org/10.7910/DVN/AOL0LP. This includes all of the relevant clinical and statistical data and also the data from the laboratory testing.

**Funding:** DSB was the sole recipient of the award to perform this study (INV-008079). The study was funded by the Bill & Melinda Gates Foundation (https://www.gatesfoundation.org/). The donor had no role role in the study design, data collection and analysis, decision to publish, or preparation of the manuscript.

**Competing interests:** The authors have declared that no competing interests exist.

of uLAM remained relatively low for people with clinical TB, which may present challenges for improving RDT sensitivity.

## Introduction

Tuberculosis (TB) remains one of the leading infectious causes of death worldwide. Most TB infections occur among HIV negative (HIV-ve) persons, but people living with HIV (PLWH) have a significantly higher individual risk of clinical TB disease and mortality [1, 2]. In 2022, an estimated 1.3 million people died including 136,000 adult ($\geq$15 years old) PLWH [2]. Many people in TB-endemic settings may have limited access to care, and improved access to TB diagnostics could improve treatment initiation and reduce mortality [3].

Lipoarabinomannan (LAM) is a pathogen biomarker with clinical utility for diagnosing TB disease [4]. LAM is a glycolipid in the Mycobacterial cell excreted in patient urine, serum, and plasma [4–6]. While dimannose-capped derivatives of the LAM glycolipid are specific to *M. tuberculosis* (MTB) and are readily detectable in LAM isolated from bacterial culture, [7–9], these structural forms of LAM are undetectable in TB patient urine samples [10, 11]. In contrast, antibodies specific for LAM derivatives with 5-methylthio-d-xylofuranose (MTX)-substituted dimannose caps can readily detect the presence of LAM in many clinical urine samples [10–12], indicating a significant difference in structure and immunoreactivity between the bacterial culture isolated forms and clinical forms of LAM. The Alere Determine TB LAM Ag (Determine, Abbott Diagnostics, USA) was developed using bacterial LAM, and is a rapid diagnostic test (RDT) approved by the World Health Organization (WHO) for use among PLWH with immune suppression and advanced disease [13–15]. However, this RDT is only approved for people with clinical TB-related symptoms who have a CD4 count of <200 cells/mm$^3$ [16]. As such, the only WHO-approved uLAM RDT has insufficient sensitivity for HIV-ve people with clinical TB. The WHO have released target product profiles (TPPs) for high priority TB diagnostic tests including a rapid biomarker-based non-sputum-based test for detecting TB [17]. Urinary LAM is a candidate for this being non-invasive and non-sputum-based biomarker with the potential to diagnose pulmonary and extrapulmonary TB in both adults and children.

Limited data exists on the concentration of urine LAM (uLAM) in HIV -ve adults with clinical TB. Prior studies of uLAM concentrations used material predominantly from mycobacterial culture which differs structurally from *in vivo* isolated LAM [18, 19]. Most clinical validation studies using RDTs for uLAM have focused on PLWH and evaluated performance against microbiological reference standards (MRS; culture, and/or GeneXpert) [20, 21]. One study describing two small cohorts from Peru and South Africa showed high specificity and varying sensitivity but however, it was underpowered for HIV-ve participants [22]. A scaled clinical evaluation of uLAM using an ultra-sensitive LAM immunoassay has yet to be performed in HIV-ve adults. The objective of the study was to accurately quantify the concentration of uLAM and evaluate clinical diagnostic accuracy among HIV-ve adults presenting with clinical TB-related symptoms using highly sensitive electrochemiluminescent (ECL) immunoassays [10]. We included adults with both presumptive pulmonary (PTB) and extrapulmonary (EPTB) TB in a high TB-burden, low HIV setting in Hanoi, Vietnam.

## Materials and methods

### Study design and participants

We conducted a prospective longitudinal cohort study of 780 adults ≥18 years of age who had clinical TB-related symptoms and presented for routine care at the National Lung Hospital (Hanoi, Vietnam) between the 4th October 2021 and 5th April 2022. Six- month follow-up visits were completed by 18th November 2022. Participants with presumptive pulmonary TB (PTB) were recruited from both the outpatient clinic and hospitalized respiratory inpatient ward using convenience sampling. Presumptive extrapulmonary (EPTB) participants were recruited from an EPTB inpatient ward exclusively using convenience sampling. Exclusion criteria included having received isoniazid preventive therapy within the previous three months, having received anti-TB treatment for more than 24 hours, or having a confirmed TB diagnosis at the time of recruitment. Details of study recruitment, eligibility and exclusion criteria, and enrollment procedures are described in the supplementary materials. Eligibility criteria were developed to represent people who may be evaluated for clinical PTB or EPTB in Vietnam (S1 and S2 Text).

### Ethical approvals

The institutional review boards (IRBs) at the National Lung Hospital (NLH, Hanoi, Vietnam; reference No: 26/21/CN-HDDD) and the Vietnam Ministry of Health (Hanoi, Vietnam; reference No: 95/CN-HDDD) approved the study and the PATH Office of Research Affairs (ORA) gave approval contingent on the NLH IRB decision. Written informed consent was obtained from each eligible participant prior to enrollment. The immunoassay testing at the PATH laboratory was determined as non-human subjects research by the PATH ORA prior to sample analysis. In the signed consent forms, each participant also agreed to the use of their urine samples beyond this study and these are available to groups developing uLAM assays. The complete study protocol can be publicly accessed at Dataverse (https://doi.org/10.7910/DVN/AOL0LP).

### Study procedures and clinical care

Following enrollment, we collected patient sociodemographic and clinical history using a standardized structured questionnaire and patient chart and laboratory record abstraction. Participants received a clinical evaluation as part of the routine standard of care. If a participant did not have a documented HIV test within the prior six-months, they were offered testing. HIV testing was offered using Determine HIV-1/2 (Abbot Diagnostics, Scarborough, ME) or Quick Test HIV 1 & 2 (Amvi Biotech, Ho Chi Minh City, Vietnam), and positive results confirmed with a second rapid test and finally an HIV ELISA (Murex HIV Ag/Ab combination, DiaSorin, Saluggia, Italy). For participants with presumptive PTB, we collected all participants' separate sputum samples for Xpert MTB/RIF Ultra and Mycobacteria Growth Indicator Tube (MGIT) liquid culture testing. If spontaneous sputum samples were unable to be provided, sputum induction was performed. For participants with presumptive EPTB, we collected non-respiratory samples (cerebrospinal fluid, pleural fluid, etc.) for MGIT and Xpert MTB/RIF Ultra testing, as clinically indicated. Urine samples were frozen at –80°C immediately following collection until the lab-based testing. At two and six months after enrollment, we conducted participant follow-up phone calls to evaluate clinical status and response to TB treatment. Clinical charts and laboratory records were abstracted to ascertain incident diagnoses, repeat hospitalization, or treatment initiation. Approximately 30 mL of urine from each participant

was collected for analysis with the electrochemiluminescent (ECL) immunoassays. No adverse events associated with index or reference testing were reported.

## Evaluation of the clinical urine specimens using immunoassays

The concentration of uLAM in the urine specimens was measured using highly sensitive ECL immunoassay [10]. We evaluated two separate anti-LAM monoclonal capture antibodies, FIND28 (Foundation for Innovative New Diagnostics, Geneva, [10]) and S4-20 (Otsuka Pharmaceuticals, Tokyo, Japan [23]), that were biotinylated (EZ-Link Sulfo-NHS-LC-Biotinylation Kit, ThermoFisher Scientific, Waltham, MA, USA). FIND28 recognizes the Ara6 epitope and any of the Man caps. S4-20 recognizes the Mtb unique MTX motif and any Man caps [9, 11]. For both capture antibodies, we used the same recombinant detector antibody, A194-01 which recognizes epitopes for uncapped Ara4 and Ara6 ± Man1 cap (Rutgers University) [9]. This was labeled with the GOLD SULFO-TAG NHS-Ester (Mesoscale Diagnostics [MSD], Rockville, MD, USA). Unbound labels (biotin or SULFO-TAG) were removed using desalting columns (40 kDa MWCO Zeba Spin, Thermo).

The biotinylated capture antibodies were coupled to U-PLEX plates (MSD) via biotin-streptavidin binding to U-PLEX linkers (MSD). Both antibody-linker conjugates were mixed with U-PLEX stop buffer (MSD) at a concentration of 0.29 µg/mL and 50 µL added to each well. Plates were incubated for 1 hour with shaking at 500 rpm to allow for the antibodies to self-assemble to their discrete and complimentary linker-binding sites to create a 2-plex immunoarray. Plates were washed 3X with 300 µL/well of 1X phosphate buffered saline + 0.05% Tween 20 (PBS-T, pH 7.5) using a ELX405R microplate washer (BioTek Instruments Inc., Winooski, VT).

uLAM quantitation was performed by adding 25 µL of Buffer 22 (MSD) to each well followed by 25 µL of unconcentrated urine sample, standard, or control [10]. Plates were incubated at room temperature with shaking for 1 hour at 500 rpm to allow binding of uLAM to the capture antibodies. Plates were washed 3 times with PBS-T and 25 µL of SULFO-TAG detection antibody (2 µg/mL) in Diluent 3 (MSD) was then added to each well with shaking for 1 hour. Plates were washed 3 times with PBS-T and filled with 150 µL of 2X read buffer T (MSD) per well. The plates were read in a MESO QuickPlex SQ 120 plate reader (MSD) and the ECLs from each individual array spot were measured and analyzed using the Discovery Workbench v4 software (MSD). The LAM used for the standard curve was derived from the *in vitro* culture of the TB strain Aoyama B (Nacalai USA, Inc., San Diego, CA), which was dissolved in 1% bovine serum albumin (BSA) in water (w/v). Duplicate seven-point serial dilutions of the TB LAM stock (40,000 pg/ml to 2.44 pg/mL) in 2% BSA/1X PBS with a negative control were used to generate calibration curves.

The relationship of ECL to LAM concentration was fitted to a four-parameter logistic (4-PL) function. The uLAM concentration in each specimen was then calculated for both antibody pairs by back-fitting the ECL data to the 4-PL fit using proprietary software from MSD. A specimen was considered uLAM positive when the ECL signal of the sample correlated with signals above the limit of detection (LOD) in the assay determined from the software. Specifically, the software will assign the LOD by calculating the sum of the mean and 2.5 times the standard deviation of the blanks, which is typical of a ligand binding assay. The LOD deviated slightly from plate to plate (S1 Table). We used the LOD established from each plate to score the test results for greater accuracy, as opposed to earlier approaches of applying universal cutoff values [19, 20]. Any value below the LOD was considered negative. In cases where sample uLAM was above the upper limit of detection, the sample was diluted in 1X PBS and

reassessed. Performers/readers of the ECL immunoassay were blinded to both clinical information and reference standard results.

## Reference standards

The microbiological reference standard (MRS) was defined as having a positive result by either Xpert Ultra or Mycobacterial culture by MGIT. This was selected as the primary reference standard due to being the gold standard for diagnosis of clinical TB in adults. For participants with presumptive PTB, a positive microbiological test result using a respiratory (sputum) specimen was required, while both a negative Xpert Ultra and MGIT culture result using a respiratory specimen was required to be considered negative. Two separate sputum samples were collected consecutively for Xpert Ultra and MGIT testing. The MRS for EPTB included tissue samples from the participant's pleura, lymph nodes, abdomen, or meninges. Due to the difficulty in using non-respiratory specimen for Xpert testing, when no Xpert testing was conducted for a non-respiratory specimen, a negative EPTB participant would consist of a negative culture using a non-respiratory specimen. The clinical reference standard (CRS) was defined as either meeting the MRS definition or receiving empiric TB treatment (with or without compatible chest X-ray) within 2 months of enrollment at the discretion of the treating clinicians. Performers/readers of the reference tests were blinded to results of the index test.

## Statistical analyses

Assuming a prevalence of 35% (proportion of screened individuals diagnosed at the NLH between July-December 2019) and test sensitivity of 80%, a sample size of 246 participants with confirmed TB (n = 703 overall) was calculated to obtain 5% precision. We report diagnostic accuracy of the ECL immunoassays using sensitivity, specificity, positive and negative predictive values, and positive and negative likelihood ratios, along with 95% confidence intervals estimated using the binomial exact (Clopper-Pearson) method. The primary reference standard was the MRS, with Xpert result alone, culture result alone and CRS reference standards presented for secondary analyses. Comparison of sensitivities or specificities between the presumptive PTB and presumptive EPTB groups were made using Fisher's exact test. Comparisons of sensitivities or specificities between antibody (Ab) pairs within the same TB group were made using McNemar test. Receiver operating characteristic (ROC) curves were calculated to evaluate the overall discriminatory ability of each antibody (Ab) pair, and statistically compared using paired ROC comparisons(R package pROC, v1.18) when comparing performance between the same participants [24]. We used unpaired ROC comparisons when comparing between presumptive PTB & presumptive EPTB individuals. 95% confidence intervals for area under the receiver operating characteristic (AUROC) curve was calculated using the DeLong test. We conducted all analyses using R version 4.1.2 [25]. All of the data generated in this study can be publicly accessed at Dataverse (https://doi.org/10.7910/DVN/AOL0LP).

## Results

### Participants recruited

We enrolled 780 participants from inpatient and outpatient wards (700 people with presumptive PTB and 80 people with presumptive EPTB). Of these, 11 participants were excluded due to the urine collection not being within 48 hours of enrollment, and 3 were excluded due to their HIV status, leaving 766 participants meeting MRS criteria for classification (693 presumptive PTB, 73 presumptive EPTB; Fig 1). From the presumptive PTB group, 1 participant could not be validated due to culture being available only on an extrapulmonary specimen

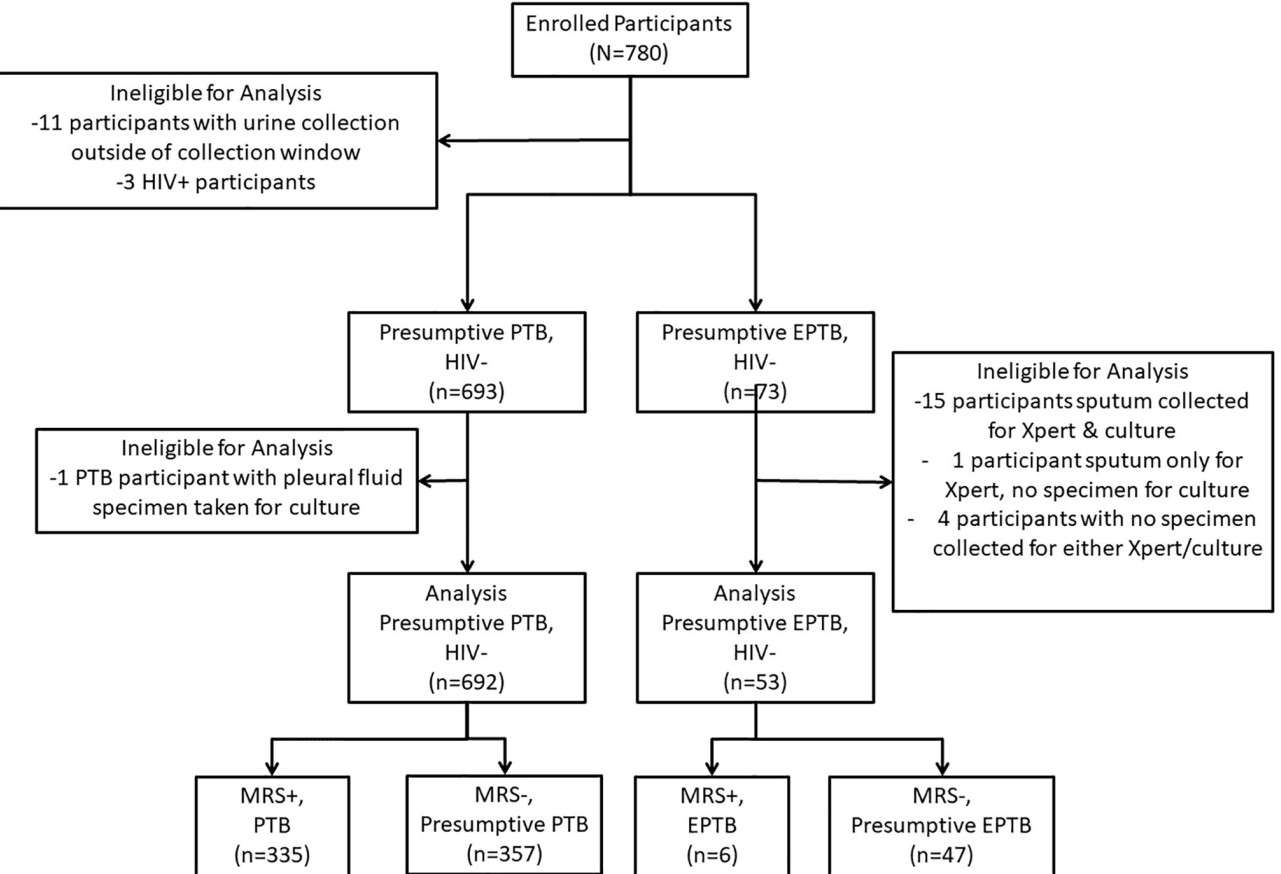

**Fig 1. A STARD accuracy flowchart of the recruitment and diagnostic classification of the study subjects from October 2021 to April 2022.** HIV, human immunodeficiency virus; Presumptive PTB, Presumptive Pulmonary TuBerculosis; Presumptive EPTB, Presumptive Extra Pulmonary TuBerculosis; Xpert, GeneXpert MTB/RIF Ultra.

(pleural fluid). From the presumptive EPTB group, 20 participants could not be validated because non-pulmonary specimens were not available for validation testing. Of these, 15 had only a pulmonary sample collection for validation, one had only a respiratory (sputum) specimen collected for Xpert testing and none for MGIT testing, and four had no sample collected for Xpert or MGIT testing. The reasons for missing samples for these four participants were that no sample fluids were available for collection from cases of presumed spinal TB, presumed intestinal TB and presumed bone TB respectively. A total of 692 presumptive PTB and 53 presumptive EPTB participants were included in our analyses.

## Participant characteristics

Overall, the participants were predominantly male (66.7%), of older age (mean = 46.7 years, SD = 16.0) and 6% self-reported a history of recent TB exposure (Table 1). The most common reported symptoms among the presumptive PTB group included prolonged cough (88.9%), weight loss (64.5%) and fatigue (67.6%), with fewer participants reporting prolonged fever (23.0%). High proportions of participants in the presumptive EPTB group reported typical TB symptoms (prolonged cough, fever, nights sweats, weight loss and fatigue). Chest X-ray (CXR) indicative of clinical TB was seen in 21.8% of presumptive-PTB participants with chest X-rays

**Table 1. Characteristics of enrolled HIV negative participants with presumptive pulmonary and extrapulmonary TB (inpatient and outpatient recruitment, N = 745).**

| | Presumptive Pulmonary TB Patients (n = 692) | Presumptive Extrapulmonary TB Patients (n = 53) | Total (N = 745) |
|---|---|---|---|
| Demographics | | | |
| Female Sex, n (%) | 231 (33.4) | 17 (32.1) | 248 (33.3) |
| Age, mean (SD) | 46.6 (15.9) | 48 (17.2) | 46.7 (16.0) |
| Hospitalization status, n (%) | | | |
| Inpatient | 178 (25.7) | 53 (100) | 231 (31.0) |
| Outpatient | 514 (74.3) | 0 (0) | 514 (69.0) |
| Symptom presentation, n (%) | | | |
| Cough | 615 (88.9) | 44 (83) | 659 (88.5) |
| Fever | 159 (23.0) | 43 (81.1) | 202 (27.1) |
| Night sweats | 159 (23.0) | 42 (79.2) | 201 (27.0) |
| Weight Loss | 446 (64.5) | 52 (98.1) | 498 (66.8) |
| Fatigue | 468 (67.6) | 53 (100) | 521 (69.9) |
| Recent household exposure*, n (%) | | | |
| Yes | 43 (6.2) | 2 (3.8) | 45 (6.0) |
| No | 641 (92.6) | 51 (96.2) | 692 (92.9) |
| Don't Know | 8 (1.2) | 0 (0) | 8 (1.1) |
| Chest X-Ray Indicative of TB**, n (%) | 97 (21.8) | 4 (7.5) | 101 (20.3) |
| AFB smear positive, n (%) | 131 (18.9) | 0 (0) | 131 (17.6) |
| AFB smear positivity, n (%) Scanty | 12 (9.1) | 0 (0) | 12 (9.1) |
| 1+ | 65 (49.6) | 0 (0) | 65 (49.6) |
| 2+ | 32 (24.4) | 0 (0) | 32 (24.4) |
| 3+ | 22 (16.8) | 0 (0) | 22 (16.8) |
| Xpert Ultra positive***, n (%) | 300 (43.3) | 5 (22.7) | 305 (42.7) |
| Rif resistance detected, n (%) | 12 (4.0) | 0 (0) | 12 (1.6) |
| MGIT liquid culture positivity, n (%) | | | |
| Positive | 280 (40.5) | 3 (5.7) | 283 (37.9) |
| NTM | 35 (5.1) | 0 (0) | 35 (4.7) |
| Negative | 375 (54.2) | 50 (94.3) | 425 (50.3) |
| Specimen contamination/not ordered | 2 (0.3) | 0 (0) | 2 (0.3) |
| Microbiological Reference Standard, n (%) | 335 (48.5) | 6 (11.3) | 341 (45.8) |
| Xpert Ultra positive***, n (%) | 300 (43.4) | 5 (22.7) | 305 (42.7) |
| Culture positive****, n (%) | 280 (42.7) | 3 (5.7) | 283 (40.0) |
| Clinical Reference Standard, n (%) | 414 (59.8) | 33 (62.3) | 447 (60.0) |

\* Captured by question "Currently living with someone with confirmed TB diagnosis?";

\*\*496 participants with available CXR results (444 presumptive PTB, 52 presumptive EPTB);

\*\*\*22 presumptive EPTB participants with non-sputum samples; denominator for presumptive PTB cohort is participants with a pulmonary specimen collected for testing, while for presumptive EPTB group is participants with a non-pulmonary specimen collected;

\*\*\*\*Participants with inadequate specimen, NTM or not ordered set to missing for culture only reference standard. Abbreviations: AFB, Acid Fast Bacilli; MGIT, mycobacterial growth incubation tube;

and 7.5% of presumptive EPTB participants. The majority (81.1%) of participants with presumptive PTB were acid fast bacilli (AFB) smear negative, and among participants who were AFB smear positive, nearly half had a bacterial load graded as scanty or 1+. Positivity by MGIT liquid culture was 40.5% in the presumptive PTB group but only 5.7% in the presumptive EPTB group. Xpert Ultra positivity was 43.3% in participants with presumptive PTB, and

22.7% (5/22) in the subset of presumptive EPTB participants with Xpert Ultra. Overall, 335 (48.5%) of presumptive PTB and just 6 (11.3%) presumptive EPTB participants were microbiologically confirmed as TB positive. Using the CRS, 414 (59.8%) of presumptive PTB and 33 (62.3%) of presumptive EPTB participants were considered TB positive. MGIT also identified 35 participants as having an infection with nontuberculous mycobacteria (NTM).

## Urinary LAM concentrations

The uLAM concentrations derived from using the FIND 28:A194-01 (F/A) antibody (Ab) pair detected uLAM in a greater number of participants in both presumptive PTB and presumptive EPTB cohorts as compared to the S4-20:A194-01 (S/A) pair (Table 2). A similar proportion of participants with presumptive PTB and presumptive EPTB had uLAM detected with F/A (30.6% vs 34.0%, $p$ = 0.643) as S4-20/A194-01 (S/A; 20.1% vs 15.1%, $p$ = 0.475) respectively when not stratifying by MRS status. When combining the two cohorts, F/A detected positivity was 30.9% (n = 230) while S/A was 19.7% (n = 147). Among participants with uLAM detected by F/A, the median estimated uLAM concentration was similar for both participants with presumptive PTB (79.0 pg/mL, (interquartile range [IQR] 31.8–290.5) and presumptive EPTB (76.5 pg/mL, IQR 34.3–190.0). When stratifying by MRS status, median estimated uLAM concentration appeared higher for MRS-positive participants from the presumptive PTB group compared to MRS-negative participants from the presumptive PTB group(110.0 pg/mL vs. 45.0 pg/mL), higher among MRS-negative compared to MRS-positive participants in the presumptive EPTB group(44.0 pg/mL vs 77.0 pg/mL). Among participants with uLAM detected in their urine using the S/A pair, the median estimated uLAM concentration among the presumptive PTB group was 55.0 pg/mL, (IQR 27.0–139.5), and 36.0 pg/mL (IQR 29.0–46.3) for the presumptive EPTB group. When stratifying by MRS status, median estimated uLAM concentration appeared higher for MRS-positive participants from the presumptive PTB group (57.0 pg/mL vs. 21.0 pg/mL) compared to MRS-negative participants, as well as for participants in the presumptive EPTB group (50.5 pg/mL vs. 33.0 pg/mL) respectively. When viewing the distribution of estimated uLAM concentrations for the same participants using the two antibody pairs, the F/A pair appears to estimate a higher concentration of 78.0 pg/mL as compared to the S4-20:A194-01 Ab pair with 51.0 pg/mL (Table 2, Fig 2 and S1 Fig). Only 87 (32.9%) presumptive PTB and 5 (23.8%) presumptive EPTB participants were uLAM positive by both F/A and S/A pairs, indicating a high degree of discordant detection (S1 Fig).

## Diagnostic performance of S/A and F/A antibody Pairs

When evaluating the diagnostic performance of uLAM detection using the S/A pair, the overall sensitivity was low in this HIV-ve population. A similar proportion of microbiologically

**Table 2. ECL test results of study participants using S/A and F/A Ab pairs (N = 745).**

| ECL Test Results | Presumptive PTB (N = 692) | | Presumptive EPTB (N = 53) | | Pooled (N = 745) |
|---|---|---|---|---|---|
| | MRS + (n = 335) | MRS–(n = 357) | MRS + (n = 6) | MRS–(n = 47) | |
| S/A positive result–N (%) | 130 (38.8) | 9 (2.5) | 2 (33.3) | 6 (12.8) | 147 (19.7) |
| S/A Median uLAM conc.–pg/mL (IQR) | 57.0 (29.0–150.0) | 21.0 (17.0–26.0) | 50.5 (50.25–50.75) | 33.0 (23.0–37.0) | 51.0 (27.0–125.0) |
| F/A Positive result–N (%) | 137 (40.9) | 75 (21.0) | 3 (50.0) | 15 (31.9) | 230 (30.9) |
| F/A Median uLAM conc.–pg/mL (IQR) | 110.0 (36.0–374.0)) | 45.0 (29.5–104.0) | 44.0 (30.5–93.0)) | 33.0 (23.0–37.0) | 78.0 (31.2–284.0) |

The number of S/A and F/A positive results for both the presumptive PTB & EPTB group, and the pooled data with the median uLAM concentration (pg/mL) among participants with detected uLAM. Abbreviations: PTB, pulmonary TB; EPTB, extrapulmonary TB; MRS, microbiological reference standard; ECL, electrochemiluminescent; S/A, S4-20/A194-01 Ab pair; F/A, FIND28/A194-01 Ab pair. IQR, interquartile range.

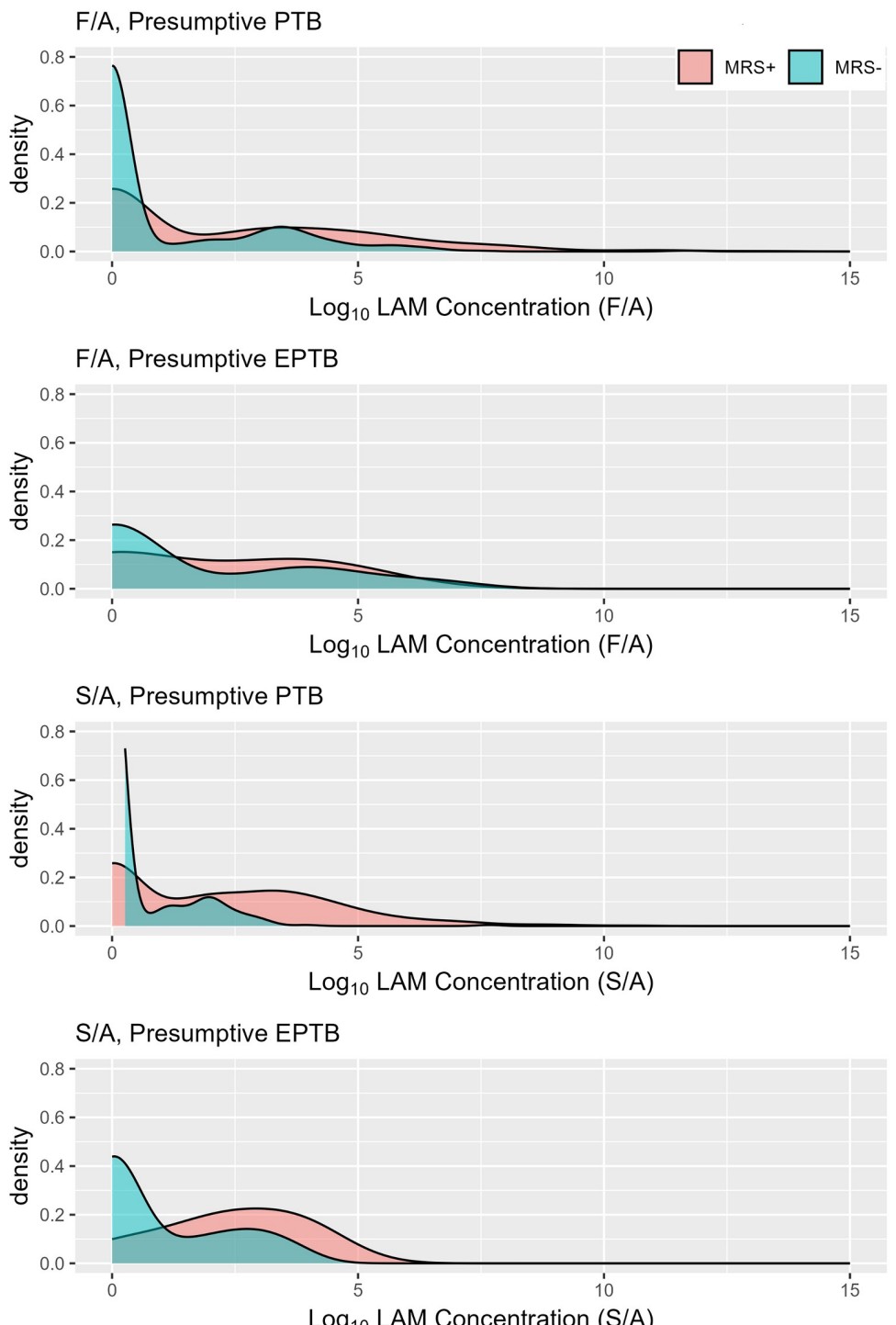

**Fig 2. The range of uLAM concentrations quantitated by the ECL immunoassay using antibody pairs S/A and F/A, stratified by MRS status and cohort (N = 745).** The uLAM concentrations as measured using the S/A and F/A antibody pairs in both MRS positive (red) and MRS negative (blue), stratified by cohort status. The data was scored as both the number of positive results observed and with the median concentration of uLAM determined from the uLAM positive samples of all positive results using the ECL assays.

confirmed participants were correctly identified in presumptive PTB (sensitivity: 39%, 95% CI: 34–44%) and presumptive EPTB (sensitivity: 33%, 95% CI: 4–78%) participants (Table 3). However, true negatives were more accurately identified in the presumptive PTB (specificity: 97%, 95% CI: 95–99%) compared to presumptive EPTB (specificity: 87%, 95% CI: 74–95%) groups. When using the CRS as the reference standard instead, test sensitivity was 32% (95% CI: 28–37%) in the presumptive PTB and 24% (95% CI: 11–42%) in presumptive EPTB group, and specificity was similar in both at 98%-100%. When the immunoassay data for S/A and F/A was pooled (uLAM positive using either antibody pair) and also compared to the MRS, Xpert Ultra, MGIT and CRS data for both presumptive PTB (S1 Table) and presumptive EPTB (S3 Table) participants gave greater sensitivity but with a concomitant loss in specificity. The sensitivity in the presumptive PTB group was 55% (95% CI: 0.49–0.60%) and the specificity was 78% (95% CI: 0.73–0.82%). In the presumptive EPTB group, sensitivity was similar at 50% (95% CI: 0.12–0.88%) and with a specificity of 62% (95% CI: 0.46–0.75%).

When evaluating the diagnostic performance of uLAM detection using the F/A pair, sensitivity was 41% (95% CI: 36–46%) in the presumptive PTB, and 50% (95% CI: 12–88%) in the presumptive PTB group (*p* = 0.693) (Table 3). Specificity was higher using F/A in the presumptive PTB (Specificity: 79%, 95% CI: 74–83%) compared to presumptive EPTB group (Specificity: 68%, 95% CI: 53–81%, *p* = 0.096). When comparing performance using the CRS as the reference standard in the presumptive PTB and presumptive EPTB groups, both sensitivity (37% vs 39%) and specificity were similar (79% vs 75%).

When assessing performance between the S/A and F/A Ab pairs, their sensitivity was similar in the presumptive PTB group (*p* = 0.356). However, the S/A pair had significantly higher specificity (97% vs 79%, *p<0.0001*) and PPV (94% vs 65%, *p<0.0001*). Similar diagnostic performances were observed between both Ab pairs when using CRS as the reference standard. However, specificity was higher for the S/A than for F/A pair (87% vs 68%, *p* = 0.039) in the presumptive EPTB group. When comparing the discriminatory ability at varying estimated concentration thresholds, the S/A pair showed greater discriminatory ability compared to F/A pair in the presumptive PTB (area under the curve [AUC] = 0.744 vs 0.628, *p<0.0001*) and presumptive EPTB group (AUC = 0.755 vs 0.5339, *p* = 0.045) (Table 3, Fig 3).

## Discussion

We have completed the largest study to date assessing the diagnostic performance of a highly sensitive immunoassay platform to detect the levels of uLAM from urine in a HIV-ve cohort presenting with presumptive clinical TB. In HIV-ve adults with either presumptive clinical PTB or EPTB, uLAM was detectable in less than half of participants, with low concentrations for the majority of positive participants. The F/A pair had slightly higher sensitivity than the S/A pair (41% vs. 39%) but also had a much lower specificity (79% vs. 97%). The majority of PTB patients were sputum smear-negative (18.9% smear positivity), suggesting that a large proportion of participants in this study had paucibacillary TB, potentially resulting in less availability of LAM at the site of infection and urine. Other studies have also noted significant differences in uLAM levels based on the geographical sites, highlighting that different TB lineages may affect the levels of uLAM observed in patient urine [15, 19]. We also investigated the performance of the immunoassays with urine collected from either pulmonary or EPTB patients. Though no significant differences were observed between these groups with either antibody pair, it should be noted the size of the presumptive EPTB group (N = 53, MRS TB positive 6) was much smaller than the presumptive PTB group (N = 692, MRS TB positive 335) and may not have been powered to detect a significant difference of smaller magnitude.

**Table 3. The diagnostic sensitivity and specificity of the ECL S/A and & F/A immunoassays in the presumptive PTB and EPTB groups as compared to the microbiologic reference standards (MRS), Xpert, MGIT, and the clinical reference standard (CRS) used in the diagnosis of clinical TB.**

| Reference Standard | N | Positive by reference standard | | Negative by reference standard | | PPV (95% CI) | NPV (95% CI) | PLR (95% CI) | NLR (95% CI) | AUROC (95% CI) |
|---|---|---|---|---|---|---|---|---|---|---|
| | | TP/TP+FN | Sensitivity (95% CI) | TN/TN+FP | Specificity (95% CI) | | | | | |
| **S/A Presumptive PTB** | | | | | | | | | | |
| MRS | 692 | 130/335 | 0.39 (0.34, 0.44) | 348/357 | 0.97 (0.95, 0.99) | 0.94 (0.88, 0.97) | 0.63 (0.59, 0.67) | 15.4 (7.96, 29.75) | 0.63 (0.58, 0.68) | 0.74 (0.71, 0.78) |
| Xpert Ultra | 692 | 123/300 | 0.41 (0.35, 0.47) | 376/392 | 0.96 (0.93, 0.98) | 0.88 (0.82, 0.93) | 0.68 (0.64, 0.72) | 15.4 (7.96, 29.75) | 0.62 (0.56, 0.68) | 0.75 (0.71, 0.79) |
| MGIT$ | 655 | 112/280 | 0.40 (0.34, 0.46) | 356/375 | 0.95 (0.92, 0.97) | 0.95 (0.92, 0.97) | 0.68 (0.64, 0.72) | 7.89 (4.98, 12.52) | 0.63 (0.57, 0.70) | 0.74 (0.70, 0.77) |
| CRS | 692 | 133/414 | 0.32 (0.28, 0.37) | 272/278 | 0.98 (0.95, 0.99) | 0.96 (0.91, 0.98) | 0.49 (0.45, 0.53) | 14.88 (6.66, 33.25) | 0.69 (0.65, 0.74) | 0.71 (0.68, 0.74) |
| **S/A Presumptive EPTB** | | | | | | | | | | |
| MRS | 53 | 2/6 | 0.33 (0.04, 0.78) | 41/47 | 0.87 (0.74, 0.95) | 0.25 (0.03, 0.65) | 0.91 (0.79, 0.98) | 2.61 (0.67, 10.13) | 0.76 (0.43, 1.36) | 0.76 (0.53, 0.98) |
| Xpert Ultra* | 22 | 2/5 | 0.40 (0.05, 0.85) | 11/17 | 0.65 (0.38, 0.86) | 0.25 (0.03, 0.65) | 0.79 (0.49, 0.95) | 1.13 (0.32, 3.96) | 0.93 (0.42, 2.06) | 0.68 (0.36, 0.99) |
| MGIT | 53 | 1/3 | 0.33 (0.01, 0.91) | 43/50 | 0.86 (0.73, 0.94) | 0.12 (0, 0.53) | 0.96 (0.85, 0.99) | 2.38 (0.42, 13.59) | 0.78 (0.35, 1.74) | 0.79 (0.58, 1.00) |
| CRS | 53 | 8/33 | 0.24 (0.11, 0.42)) | 20/20 | 1.00 (0.83, 1.00) | 1.00 (0.63, 1.00) | 0.44 (0.30, 0.60) | Inf (NA, Inf) | 0.76 (0.62, 0.92) | 0.66 (0.52, 0.79) |
| **F/A Presumptive PTB** | | | | | | | | | | |
| MRS | 692 | 137/335 | 0.41 (0.36, 0.46) | 282/357 | 0.79 (0.74, 0.83) | 0.65 (0.58, 0.71) | 0.59 (0.54, 0.63) | 1.95 (1.53, 2.47) | 0.75 (0.67, 0.83) | 0.63 (0.59, 0.66) |
| Xpert Ultra | 692 | 124/300 | 0.41 (0.36, 0.47) | 304/392 | 0.78 (0.73, 0.82) | 0.58 (0.52, 0.65) | 0.63 (0.59, 0.68) | 1.84 (1.47, 2.31) | 0.76 (0.68, 0.84) | 0.62 (0.58, 0.66) |
| MGIT$ | 655 | 118/280 | 0.42 (0.36, 0.48) | 290/375 | 0.77 (0.73, 0.81) | 0.58 (0.51, 0.65) | 0.64 (0.60, 0.69) | 1.86 (1.47, 2.34) | 0.75 (0.67, 0.84) | 0.62 (0.58, 0.66) |
| CRS | 692 | 154/414 | 0.37 (0.33, 0.42) | 220/278 | 0.79 (0.74, 0.84) | 0.73 (0.66, 0.79) | 0.46 (0.41, 0.50) | 1.78 (1.37, 2.31) | 0.79 (0.72, 0.87)v | 0.60 (0.56, 0.64) |
| **F/A Presumptive EPTB** | | | | | | | | | | |
| MRS | 53 | 3/6 | 0.50 (0.12, 0.88) | 32/47 | 0.68 (0.53, 0.81) | 0.17 (0.04, 0.41) | 0.91 (0.77, 0.98) | 1.57 (0.64, 3.86) | 0.73 (0.32, 1.67) | 0.54 (0.31, 0.78) |
| Xpert Ultra* | 22 | 2/5 | 0.40 (0.05, 0.85) | 11/17 | 0.65 (0.38, 0.86) | 0.25 (0.03, 0.65) | 0.79 (0.49, 0.95) | 1.13 (0.32, 3.96) | 0.93 (0.42, 2.06) | 0.48 (0.21, 0.75) |
| MGIT$ | 53 | 2/3 | 0.67 (0.09, 0.99) | 34/50 | 0.68 (0.53, 0.80) | 0.11 (0.01, 0.35) | 0.97 (0.85, 1) | 2.08 (0.85, 5.11) | 0.49 (0.10, 2.46) | 0.63 (0.27, 0.98) |
| CRS | 53 | 13/33 | 0.39 (0.23, 0.58) | 15/20 | 0.75 (0.51, 0.91) | 0.72 (0.47, 0.90) | 0.43 (0.26, 0.61) | 1.58 (0.66, 3.76) | 0.81 (0.56, 1.17) | 0.59 (0.45, 0.73) |

The positive predictive values (PPV), negative predictive values (NPV), positive likelihood ratios (PLR) and negative likelihood ratios (NLR) were also calculated for each. Abbreviations: TP, true positive; TN, true negative; FN, false negative; FP, false positive; CI, confidence interval; S/A, S4-20/A194-01; F/A, FIND28/A194-01; PTB, pulmonary tuberculosis; MGIT, mycobacterial growth indicator tube; MRS, microbiological reference standard; CRS, clinical reference standard; EPTB, extra pulmonary tuberculosis; AUROC, Area under the receiver operating curve.

$ Participants with inadequate specimen, NTM, or not ordered set to missing for culture only reference standard.

* 22 participants had a non-sputum sample taken for Xpert Ultra testing for EPTB; participants with a sputum sample taken were excluded from analysis. The 22 shown due to having a non-sputum sample for Xpert Ultra testing are likely not a random sample of the total 53 participants in the table, and diagnostic performance estimates should be interpreted with caution.

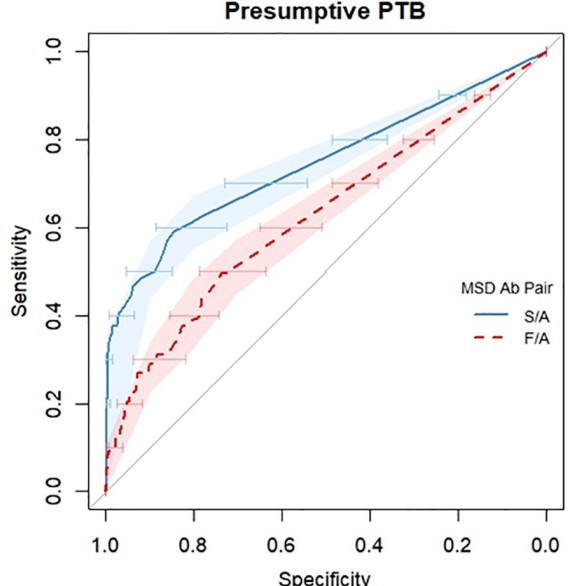
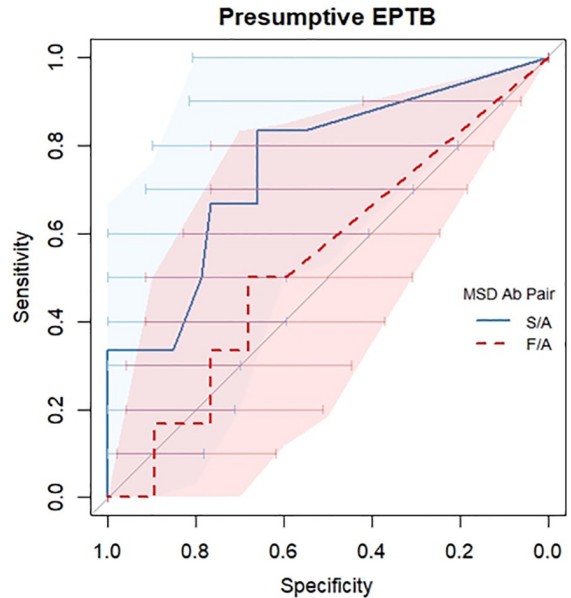

**Fig 3. Receiver operating characteristic curves including sensitivity and specificity 95% CIs comparing estimated uLAM concentrations (pg/mL) derived from the S/A and F/A Ab pairs to MRS.** ROCs for presumptive PTB (n = 692) and EPTB (n = 53) are depicted on the left & right, respectively. The ROC for S/A is shown by the solid blue line, while the ROC for F/A is shown by the dashed red line. 95% CIs for threshold sensitivity is depicted vertically by the shaded area, with 95% CI threshold specificity shown by horizontal bars. Abbreviations: CI, confidence interval; S/A, S4-20/A194-01; F/A, FIND28/A194-01; PTB, pulmonary tuberculosis; MRS, microbiological reference standard; CRS, clinical reference standard; EPTB, extra pulmonary tuberculosis.

The majority of previous cohorts screened for LAM in urine typically have a higher prevalence of HIV comorbidity and generally have higher concentrations of uLAM. The results from this study are less optimist than from other studies who reported significantly higher sensitivities when detecting uLAM in HIV-ve cohorts [26–28]. Broger et al. analyzed concentrated urine from 372 HIV-ve participants from Peru and South Africa using the same S/A assay on the MSD platform as this study, demonstrating a pooled sensitivity of 66.7% (95% CI 57.5%-74.7%) and a specificity of 98.1% (95% CI 95.6%-99.2%) [22]. In a subgroup analysis stratified by location, sensitivity was much higher in Peru (78.5%) compared with South Africa (37.5%), with specificities of 100% and 95.3% respectively [22]. This reflects that the poor sensitivity of uLAM detection using the MSD platform has been observed in similar cohorts. While we used the same platform and ECL assays as an earlier smaller scale study on an HIV-ve cohort [10], we used a different methodology for establishing the detection threshold. In the work by Sigal et al., a cut off value of 11 pg/mL was assigned to the S/A assay results [10], while Broger et al. used a cutoff of 5.4 pg/mL [22], noting that such a cutoff would not be applicable to other LAM assays. Where a cutoff allows for practicality and comparative analysis, using this ultrasensitive method for uLAM is still relatively new. Furthermore, using plate based LOD allows for a more standardized and objective measure of a given analyte, especially one as heterogenous as LAM [9, 18, 27], where there are inherent differences in the antibody pairs being assessed respective to sensitivity and specificity. Once use of a specific antibody pair is normalized, a generalizable cutoff would be expeditious in studying LAM in various populations. While the negative association of glycosuria on uLAM concentration with A194-01 and an S4-20 derivative has been noted, we did not perform urinalysis on the samples used and so a greater proportion of participants with glycosuria would influence our results [27].

Other studies have measured uLAM using different methods. One study reported the use of nanocages hosting a dye as the capture ligand in association with three detector antibodies including A194-01 and MoAB1, a recombinant derivative of S4-20 [29]. The nanocages offered a 50-fold increase in uLAM concentration and overall, the pooled diagnostic accuracy from the three antibodies was 90% sensitivity at 73.5% specificity [27, 29]. A separate study with 160 predominantly HIV-ve participants also reported very high sensitivity and specificity using the antibody pair CS-35/A194-01, but the authors acknowledge that the study focused on a predominantly TB positive cohorts (140 TB positive from 160 samples, many of which were smear 1+ or greater) which would bias the reported diagnostic accuracy due to the low number of TB negative participants included (N = 10) [30]. This study also included proteinase K pretreatment and mild heating (55˚C) which significantly improved the sensitivities of the immunoassay used. In addition, all collected samples were from Peruvian patients and we speculate that this data supports the theory highlighted by Broger et al. for variations in uLAM concentrations due to geographical differences in *Mtb* lineages.

There are several limitations to this study. First, all of the ECL testing was conducted on previously frozen and not fresh urine specimens. While banked samples are more convenient for high throughput testing, long-term freezing may diminish uLAM levels [31, 32]. While samples were maintained at -80˚C on the day of collection until the time of testing, we did not assess the time from specimen collection to freezing. To limit uLAM degradation, samples were stored at 4˚C soon after collection, then stored at -80˚C once 15 mL conical tubes had been filled. Another limitation is the power calculations for the study were based on pre-pandemic national TB prevalence rates wherein an estimated 271 confirmed TB positive participants would be identified within a cohort of 780. While this improves the precision of our sensitivity estimates, the smaller proportion of TB-negative participants could potentially affect specificity precision and other estimates. Another limitation is the inclusion of 35 participants with NTM identified by MGIT in the MRS (10-MRS positive, 25-MRS-negative), which may introduce a degree of misclassification bias to our estimates. Though excluded from the culture reference standard, 10 participants with NTM that were Xpert Ultra positive were classified as MRS-positive while 25 participants with NTM that were Xpert Ultra negative were classified as MRS-negative. Despite the high specificity of Xpert Ultra, there is still the potential of misdiagnosis with certain species of NTM at high bacterial loads [33]. Of the 25 NTM participants that were MRS-negative, 9 still had low detectable uLAM (median: 8.0 pg/mL, IQR: 7.0–11.0 pg/mL). While this does not significantly affect our estimates (as supported by similar estimates between the MRS & MGIT reference standards, Table 3), this does introduce bias.

The predominant lineage of the *Mycobacterium tuberculosis* strain may also play a role in variation of the median level of uLAM detected [15, 24]. To address this, we took a different approach by setting the detection threshold to the LOD determined on a plate-by-plate basis for measuring the presence of uLAM to reduce any bias that a uniform cut-off could introduce from potentially disparate *M. tuberculosis* lineages endemic to other study populations [10, 22]. In addition, structural variation in LAM structure associated with varying *Mtb* lineages affecting antibody binding may also explain decreased ECL sensitivity in our study [18]. We have confidence in the quality of the data generated to describe the PTB cohort based on the large number of participants and the relative ease with which diagnostic results could be applied classify a participant as TB positive or negative. However, the study was not designed to enroll a sufficient number of presumptive EPTB participants to allow statistical comparisons between the presumptive PTB & presumptive EPTB group, and this was further complicated by the difficulty in the collection and use of appropriate EPTB samples to microbiologically confirm diagnosis. Participants recruited to one group were not reclassified as having confirmed PTB or EPTB unless a positive Xpert Ultra or MGIT culture result was obtained using a

respiratory or extrapulmonary specimen, respectively. Though rare, certain participants may still have had both pulmonary and extrapulmonary forms of TB, as shown by the 7.5% of presumptive EPTB participants with a positive CXR, which may introduce further misclassification bias to our findings.

Finally, the prevalence of bacteriologically confirmed TB is higher than expected in a population of presumptive adults presenting at both outpatient and inpatient settings. It is important to note that study enrollment coincided with Vietnam's surge in SARS-CoV-2 infections associated with the spread of the Omicron variant, which could have potentially increased household risk of TB exposure and affected healthcare-seeking behavior due to mobility restrictions and increased screening at health facilities. Further, greater awareness of the risks of respiratory disease may have induced earlier care seeking in some cases and as such individuals with TB were identified earlier and in addition lack of access to care during lock downs may have led to cases of self-clearance of TB. This could potentially explain the low proportion of reported febrile illness observed in this cohort, and could potentially result in patients only seeking care with increased disease severity. However, the higher proportion of AFB smear microscopy results with scanty/1+ grading indicate that this may not be the case. Nonetheless, it is important to take this into consideration when evaluating the positive and negative predictive test values, which are influenced by underlying disease prevalence.

Biomolecules in urine samples may present challenges to sensitivity, and several studies have investigated specimen preparation methods to improve access for the antibody to its target epitope [34]. Improved methods include specimen preparation via enzymatic or inhibitor specific treatments to release sequestered uLAM from complexes or remove inhibitory compounds such as lipids or protein-uLAM complexes [35–37]. The enrichment of uLAM concentration in urine specimens has been achieved via a variety of methods using either the direct capture of uLAM via specific antibodies [38, 39], a uLAM specific chemical ligand or via proteolysis, chemical treatment and ultrafiltration [29, 40]. Others have focused on improving high sensitivity assays via platform improvement, novel antibodies, and the application of machine learning algorithms to improve diagnostic accuracy [11, 41, 42]. The application of multiple antibodies, each targeting different LAM epitopes, has also been demonstrated to increase diagnostic performance [27]. A key component to this work is to ensure that clinical samples are used early in the product development phase and that clinical evaluation should encompass multiple populations as evidenced in this and other work where the detection of uLAM in the urine of TB positive HIV-ve patients varies [11, 22, 30].

In conclusion, from a large HIV-ve cohort in Vietnam, high-sensitivity immunoassay testing for uLAM did not reach the levels noted in the TPP criteria for an effective diagnostic test for clinical TB ($\geq$68% sensitivity for smear-negative, culture positive pulmonary TB, 98% specificity versus MRS). Developing a better uLAM assay to diagnose the majority of people with subclinical & clinical TB may require novel antibodies, a urine concentration step, or both. Nonetheless, developing non-sputum based RDTs to diagnose people with clinical TB needs to remain a global health priority.

## Supporting information

**S1 Table. The diagnostic sensitivity and specificity of the pooled ECL immunoassays antibody pairs.** The presumptive PTB and presumptive EPTB groups compared to the MRS, Xpert, MGIT and the CRS were used for the diagnosis of active TB. Abbreviations: N, number; TP, true positive; TN, true negative; FN, false negative; FP, false positive; CI, confidence interval; PPV, positive predictive value; NPV, negative predictive value; PLR, positive likelihood ratio; NLR, negative likelihood ratio S/A, S4-20/A194-01; F/A, FIND28/A194-01; -PTB,

pulmonary tuberculosis; MGIT, mycobacterial growth indicator tube; MRS, microbiological reference standard; CRS, clinical reference standard; EPTB, extra pulmonary tuberculosis; N/A, not applicable. $Participants with inadequate specimen, NTM, or not ordered set to missing for culture only reference standard. £1 participant excluded due to invalid result using visual assessment criteria. * 22 participants had a non-sputum sample taken for Xpert Ultra testing; participants with a sputum sample taken were excluded from analysis. The 22 shown due to having a non-sputum sample for Xpert Ultra testing are likely not a random sample of the total 53 participants in the table, and diagnostic performance estimates should be interpreted with caution.
(DOCX)

**S2 Table. Plate lower limits of detection (LLOD) of presumptive PTB group.** The LLODs from the immunoassay plates used for uLAM quantitation using the S4-20/A194-01 immuno-assay stratified by MRS status from the presumptive PTB group (n = 692). Abbreviations: N, number; MRS, microbiological reference standard; N/A, not applicable.
(DOCX)

**S3 Table. Plate lower limits of detection (LLOD) of presumptive EPTB group.** The LLOD from the immunoassay plates used for uLAM quantitation using the S4-20/A194-01 immuno-assay stratified by MRS status from the presumptive EPTB group(N = 53). Abbreviations: N, number; MRS, microbiological reference standard; N/A, not applicable.
(DOCX)

**S1 Fig. Urine log LAM concentration by the ECL immunoassay using antibody pairs S/A and F/A (N = 745).** A comparison of the distribution of estimated uLAM concentrations for the same participants using S/A and F/A the two antibody pairs.
(TIF)

**S1 Text. Participant inclusion and exclusion criteria.**
(DOCX)

**S2 Text. Study procedures.**
(DOCX)

**S1 Checklist. Inclusivity in global research.**
(DOCX)

**S2 Checklist. STARD checklist.** The checklist of standards met for reporting diagnostic accuracy studies.
(DOCX)

## Acknowledgments

The project team would like to thank each of the participants in this study for their willingness to enroll, provide urine samples to support this work, and to permit the use of their test data. We would also like to acknowledge the staff of the National Lung Hospital of Hanoi for their dedication and commitment to the implementation and operation of this project.

### Funding statement

This work was supported, in whole or in part, by the Bill & Melinda Gates Foundation Grant #OPP1208704. The conclusions and opinions expressed in this work are those of the author(s) alone and shall not be attributed to the Foundation. Under the grant conditions of the

Foundation, a Creative Commons Attribution 4.0 License has already been assigned to the Author Accepted Manuscript version that might arise from this submission. Please note works submitted as a preprint have not undergone a peer review process.

## Author Contributions

**Conceptualization:** Nguyen B. Hoa, Mark Fajans, Hung Nguyen Van, Bao Vu Ngoc, Nhung Nguyen Viet, Helen L. Storey, Paul K. Drain, David S. Boyle.

**Data curation:** Nguyen B. Hoa, Mark Fajans, Bao Vu Ngoc, Hoa Nguyen Thi, Lien Tran Thi Huong, Diep Bui Ngoc, An Tran Khanh, Katherine K. Thomas, Helen L. Storey, Paul K. Drain.

**Formal analysis:** Mark Fajans, Diep Bui Ngoc, Lorraine Lillis, Katherine K. Thomas, Roger B. Peck, Jason L. Cantera, Eileen Murphy, Helen L. Storey, Paul K. Drain, David S. Boyle.

**Funding acquisition:** David S. Boyle.

**Investigation:** Mark Fajans, Bao Vu Ngoc, An Tran Khanh, Lorraine Lillis, Marcos Perez, Roger B. Peck, Jason L. Cantera, Eileen Murphy, Paul K. Drain, David S. Boyle.

**Methodology:** Mark Fajans, Bao Vu Ngoc, Lorraine Lillis, Marcos Perez, Roger B. Peck, Jason L. Cantera, Eileen Murphy, Helen L. Storey, Abraham Pinter, Paul K. Drain, David S. Boyle.

**Project administration:** Nguyen B. Hoa, Hung Nguyen Van, Bao Vu Ngoc, Nhung Nguyen Viet, Lien Tran Thi Huong, Dung Tran Minh, Hai Nguyen Viet, Eileen Murphy, Olivia R. Halas, David S. Boyle.

**Resources:** Dung Tran Minh, Abraham Pinter, Morten Ruhwald, David S. Boyle.

**Supervision:** Nguyen B. Hoa, Bao Vu Ngoc, Hoa Nguyen Thi, Cuong Nguyen Kim, Trinh Ha Thi Tuyet, Tri Nguyen Huu, Katherine K. Thomas, Eileen Murphy, Paul K. Drain, David S. Boyle.

**Validation:** Katherine K. Thomas, Paul K. Drain, David S. Boyle.

**Visualization:** Mark Fajans.

**Writing – original draft:** Mark Fajans, Lorraine Lillis, Katherine K. Thomas, Paul K. Drain, David S. Boyle.

**Writing – review & editing:** Nguyen B. Hoa, Mark Fajans, Bao Vu Ngoc, Lien Tran Thi Huong, Katherine K. Thomas, Roger B. Peck, Jason L. Cantera, Eileen Murphy, Helen L. Storey, Abraham Pinter, Morten Ruhwald, Paul K. Drain, David S. Boyle.

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
