## [Editor Report · Decision Letter 0]

20 Dec 2023

PGPH-D-23-02171

Urine lipoarabinomannan concentrations among HIV-uninfected adults with pulmonary or extrapulmonary tuberculosis disease in Vietnam

Dear Dr. Boyle,

Many thanks for transferring this manuscript to PLOS Global Public Health. I would like to invite you to revise your manuscript, with a point-by-point rebuttal to the original decision letter comments from PLOS Medicine. I have copied these below. We will then be in a position to evaluate further for consideration of publication in PLOS Global Public Health

We look forward to receiving your revised manuscript.

Kind regards,

Rishi Kumar Gupta

Academic Editor

Journal Requirements:

1. We do not publish any copyright or trademark symbols that usually accompany proprietary names, eg  ©, ®, ™  (e.g. next to drug or reagent names). Please remove all instances of trademark/copyright symbols throughout the text, including ™.

2. Please include a complete copy of PLOS’ questionnaire on inclusivity in global research in your revised manuscript. Our policy for research in this area aims to improve transparency in the reporting of research performed outside of researchers’ own country or community. The policy applies to researchers who have travelled to a different country to conduct research, research with Indigenous populations or their lands, and research on cultural artefacts. The questionnaire can also be requested at the journal’s discretion for any other submissions, even if these conditions are not met.  Please find more information on the policy and a link to download a blank copy of the questionnaire here: https://journals.plos.org/globalpublichealth/s/best-practices-in-research-reporting. Please upload a completed version of your questionnaire as Supporting Information when you resubmit your manuscript.

**PlosMedicine decision letters notes**

Comments from the academic editor:

I am less concerned about language and clarity than whether there is anything very new here. They are correct that not many other studies have explored urine LAM in HIV negative patients but largely because the original findings were fairly dismal and did not seem to warrant a big study. So the results are not really a big surprise or anything anyone would not have expected. While I think the study is reasonably well done, I don't think it reaches the level of interest that one might expect at PLoS Med. If they have found it worked better, that would have been something to report.

Reviewer Notes:

Reviewer #1: Hoa NB et al. studied the use of uLAM levels in HIV-/TB patients as an RDT (using electrochemiluminescent immunoassays), and obtained data are compared to microbiological reference standards. The authors concluded that HIV-negative TB adults (Patients from Vietnam) show low concentrations of uLAM and suggest challenges are imperative for developing a more sensitive rapid uLAM test. The problem approached in this study is relevant worldwide because the search for alternatives for the diagnosis and follow-up of TB patients is necessary to stop TB. Although the topic is very pertinent and the sample size used in the study is very good, this manuscript has several major and minor comments that must be solved before being considered for publication. Plos Medicine is a journal with high impact, and the authors send a careless manuscript. I suggest that authors consider the below comments, take care of details in the manuscript and submit again.

Major comments

1. The abstract is not clear; several expressions should be defined before use. It should be rewritten.

2. The introduction section should be rewritten because is not clear, for instance, 1) "LAM is a glycolipid in the Mycobacterial cell…" but authors do not clarify if they will search manLAM (from virulent mycobacteria) or only LAM which could be present in mycobacteria virulent and virulent; 2) TB is caused by M. tuberculosis, and they should explain because if they search LAM, it is also present in M. bovis and it can induce TB under specific conditions. Thus, it is not clear.

3. Authors wrote as Exclusion criteria "…having a confirmed TB diagnosis at the time of recruitment" (line 85). So, this means that if the patients have a TB diagnosis was not included even if they do not receive yet an anti-TB therapy?

4. Authors wrote, "…monoclonal capture antibodies.." (line 115). Do they are anti-LAM antibodies? And the same questions to capture antibodies (line 117). I think they are anti-LAM, but authors must indicate it clearly in their methods because readers should not assume.

5. The Methods section has several abbreviations that are not indicated the mean.

6. Figure 1: What means the "ev" in HIV+ev and HIV-ve? I asked the same question to MRS.

7. Do PTBW and EPTBW patients presented essential differences in the blood cell account (mainly immune cells)? This data can be obtained from the data from clinical laboratory tests.

8. What could be an alternative to improve the sensitivity of the proposed test?

9. Limitations of the study should be indicated.

10. Could this technique be used to quantify LAM in serum? Several tests also are used to evaluate LAM in HIV-, here are discussed several doi: Front Microbiol. 2021 Apr 15;12:638047. doi: 10.3389/fmicb.2021.638047.

11. What advantages has this technique compared with other reports to quantify LAM in urine? For instance, Sci Rep. 2023 Jul 18;13(1):11560. doi: 10.1038/s41598-023-38740-3.; PLoS One. 2023 Jul 14;18(7):e0288605. doi: 10.1371/journal.pone.0288605. ACS Nano. 2023 Apr 11;17(7):6998-7006. doi: 10.1021/acsnano.3c01374.; J Clin Lab Anal. 2022 Feb;36(2):e24238. doi: 10.1002/jcla.24238., among others.

Minor comments

12. What means EPTB (line 35)? It is not indicated in the abstract.

13. What is S4-20/A194-01 (S/A) (line 36)?

14. What is FIND28/A194-01 (F/A) (line 37)?

15. The RDT abbreviation used in line 60 has not means, although it was indicated in the abstract, in the manuscript should be indicated the first time that you use it.

16. Tables also need an explanation of used abbreviations.

17. Language and grammatic need a deep revision.

Reviewer #2: This is a well-conducted study on urine lipoarabinomannan concentrations among HIV- uninfected adults with pulmonary or extrapulmonary tuberculosis disease in Vietnam. The study design, datasets, statistical methods and analyses, and presentation (tables and figures) and interpretation of the results are mostly adequate and of a good standard. Only a couple of minor issues needing attention.

1) Table 3 is a key table summarising the results of test performance against various reference standards. However, could authors please add AUROC with 95% CI for the overall performance for each test?

2) Performance comparison is a very important part of the study. The authors have done this in Page 14 and figure 3. However, it would be good to add a Table 4 to summarise these results with all the AUROC, 95% CI and P-values so that it's informative and easy to follow.

Reviewer #3: The manuscript "Urine lipoarabinomannan concentrations among HIV-uninfected adults with pulmonary or extrapulmonary TB disease in Vietnam." is an excellent study. The manuscript is well wriiten with appropriate controls and a good sample size. The utility of the assay in detecting u-LAM concentrations is unclear, however, it makes an interesting study, using two different capture antibodies and keeping the detection antibody same, to be used as a diagnostic tool to ascertain the presence or absence of LAM in both pulmonary and extra-pulmonary TB patients, although the number of EPTB sample size is limited. It would be pragmatic to see what effects the urine concentration or any form of pretreatment before the immunoassay is performed would have on the outcome of the study since many published reports have shown that pretreatment helps with the sensitivity of the assay. The authors also highlighted an important point about the geographical origin of samples having an effect on the differences in u-LAM levels. May be a study testing sample cohorts from various TB endemic regions across the world would address that more rigorously. The authors also specified that there were some 35 participants that had NTM co-infection. Did the co-infection have any effect on the u-LAM levels in those 35 subjects when compared to the TB smear gradation or compared to the remaining cohort u-Lam levels.

Best wishes,

Dr Rishi K Gupta

Academic Editor
---

## [Decision Letter · Decision Letter 1]

30 May 2024

PGPH-D-23-02171R1

Urine lipoarabinomannan concentrations among HIV-uninfected adults with pulmonary or extrapulmonary tuberculosis disease in Vietnam

Dear Dr. Boyle,

Thank you for submitting your manuscript to PLOS Global Public Health. After careful consideration, we feel that it has merit but does not fully meet PLOS Global Public Health’s publication criteria as it currently stands. Therefore, we invite you to submit a revised version of the manuscript that addresses the points raised during the review process.

**Associate Editorial comments:**

Thank you to the authors for doing a thorough job of addressing the initial reviewers’ critique. The revised article has been reviewed by two peer reviewers. I have provided Editorial comments summarizing the key recommendations for further revisions from both Editorial and Peer Review, along with the full Reviewer comments below. Please consider the summarized Editorial recommendations as essential, and the further full Peer Review comments as for consideration.

**Editorial recommendations:**

I agree with the Reviewer that the “PR-PTBW” and “PR-EPTBW” ward terminology is confusing, particularly because presumptive PTB participants were mostly recruited in an outpatient setting. Suggest rewording throughout as "presumptive PTB" and "presumptive EPTB", without further abbreviation.

In the abstract, median uLAM concentrations among the MRS-positives are difficult to interpret with no context. Suggest either removing from abstract, or include comparisons to MRS negative. An indication of how cut-offs were derived would also be helpful in the abstract.

In the introduction, lines 56-60 are difficult to follow. Suggest rephrasing for a general audience.

In the methods, some brief details of how demographic and clinical histories were captured would be helpful, e.g. was there a structured participant questionnaire or case record form?

Please also clarify whether sputum induction was performed, if spontaneous samples were not possible? And how were trace Ultra results handled?

A STARD checklist is strongly recommended.

In the results, please rephrase line 185 as it currently suggests there were “766 participants meetings MRS criteria”.

There is a lot of duplicate information in the Tables. Table 2 could be removed as most of these data are in either Table 3 or Figure 2 (median uLAM concentrations among MRS positives and negatives could be added to Figure 2). I appreciate Table 4 was added in response to a previous Reviewer comment, but since all of this information is now in Table 3, Table 4 could also be removed to avoid duplication and streamline the manuscript.

Line 260 suggests that the S/A and F/A data were “pooled”, but it is not clear what this means

In line 282, it would be clearer to state “Specificity was higher using F/A…”

Please avoid overinterpreting point estimates when sample sizes are very small, as highlighted by Reviewer 2 in line 290.

We look forward to receiving your revised manuscript.

Kind regards,

Rishi Kumar Gupta

Academic Editor

Journal Requirements:

Additional Editor Comments (if provided):

Reviewers' comments:

Reviewer's Responses to Questions

**Comments to the Author**

1. If the authors have adequately addressed your comments raised in a previous round of review and you feel that this manuscript is now acceptable for publication, you may indicate that here to bypass the “Comments to the Author” section, enter your conflict of interest statement in the “Confidential to Editor” section, and submit your "Accept" recommendation.

Reviewer #1: (No Response)

Reviewer #2: (No Response)

2. Does this manuscript meet PLOS Global Public Health’s publication criteria? Is the manuscript technically sound, and do the data support the conclusions? The manuscript must describe methodologically and ethically rigorous research with conclusions that are appropriately drawn based on the data presented.

Reviewer #1: (No Response)

Reviewer #2: Yes

3. Has the statistical analysis been performed appropriately and rigorously?

Reviewer #1: (No Response)

Reviewer #2: Yes

4. Have the authors made all data underlying the findings in their manuscript fully available (please refer to the Data Availability Statement at the start of the manuscript PDF file)?

Reviewer #1: (No Response)

Reviewer #2: No

5. Is the manuscript presented in an intelligible fashion and written in standard English?

Reviewer #1: (No Response)

Reviewer #2: Yes

6. Review Comments to the Author

Reviewer #1: Nguyen B. Hoa, Mark Fajans, and their team conducted a comprehensive study to investigate the use of uLAM levels in patients with HIV and TB using electrochemiluminescent immunoassays. They meticulously compared the results of uLAM to microbiological reference standards to determine the sensitivity of this immunoassay, which utilized only FIND28 and S4-20 as capture antibodies. The authors found that their cohort of TB patients from Vietnam had low concentrations of uLAM, and identified some factors associated with these results. This thorough approach, coupled with a sufficient sample size and statistical analysis, instills confidence in the validity of the study. However, the paper has several limitations that need to be addressed and clarified before it can be accepted.

Minor corrections

Inconsistencies:

1. The authors compare the characteristics of enrolled seronegative participants with pulmonary and extrapulmonary presumptions with respect to the values in Table 1; however, the ideas are not supported by the statistical p-values obtained. The authors should show the p-values in the table and indicate the statistical analysis performed in the footnote.

2. The use of "HIV-ve" or "MRS+ev" throughout the manuscript. although in some legends it is observed, I think it would be clearer to leave it in the main text at the level of the introduction.

3. Previously, it was mentioned that the use of uLAM means the detection of LAM in urine. The text is repetitive, so write a paragraph like “levels of uLAM from urine.”

Grammatical:

Line 28: Replace “,but there…” with “.However, there is...”

Line 36: change “,while,” with “. In contrast,”

Line 36: Replace “for MRS positive” with “MRS-positive”

Line 37: Add “the” before “FIND28/A194-01”

Line 40: Add “a” before “specificity of 79%”

Line 49: What is the meaning of “HIV-ve persons”? In the write is not clear where is defined.

Line 50: Replace “people with TB disease were either not diagnosed” with “TB people were either undiagnosed…”

Line 54: Replace “that has clinical utility for diagnosing TB disease” with “with clinical utility for diagnosing TB.”

Line 55: Delete “that is also”

Line 56: Replace “, and while..” with “. While..”

Line 70: Add comma after “culture” and delete “/or”

Line 71: Change “results, but were underpowered..” with “. However, it was underpowered..”

Line 72: Change “has not been performed..” with “has yet to be performed..”

Line 77: Change “of” with “in”

Line 81: change “adults 18 years of age or older” with “adults > 18 years old”

Line 83: Change 6 month with “Six-month”

Line 88: Change 3 month with “three months”

Line 89: Add “the” before “time”

Line 91: Change “be representative of” with “represent people”

Line 105: Change 6 month with “Six-month”

Line 108: Replace “For PR-PTBW, we collected separate sputum samples for Xpert MTB/RIF Ultra and Mycobacteria Growth Indicator Tube (MGIT) liquid culture testing for all participants.” With “For PR-PTBW, we collected all participants' separate sputum samples for Xpert MTB/RIF Ultra and Mycobacteria Growth Indicator Tube (MGIT) liquid culture testing.”

Line 112: Delete “which were”

Line 112: Change 2 and 6 months with “two and six months”

Line 114: add “s” at the end of chart.

Line 115 delete “s” at the end of mL.

Line 197: a grammatical mistake “PResumed"

Additional references:

Line 55: Change reference No.4 by https://doi.org/10.3389/fmicb.2021.638047

Line 56: LAM can be found in serum or plasma. The authors should mention it. https://doi.org/10.3389/fmicb.2021.638047

Major corrections

Inaccuracies:

1. The authors comment that the low sensitivity and specificity of the technique to identify uLAM in HIV-uninfected adults with presumed PTB or EPTB disease is due to factors that are undoubtedly correct. However, the possibility of changes in the structure is not considered in LAM, as the antibodies used do not recognize it. The authors should consider this fact as a point of discussion.

2. It has been reported that storage time can affect the identification and quantification of Lipoarabinomannan (LAM) in urine. This could explain why LAM was not identified in over half of the samples tested under the proposed assay. 10.1183/23120541.00115-2018

3. The authors propose that utilizing sample pretreatment methods can aid in releasing LAM, which is often trapped in proteins found in urine. To further support this point, it may be beneficial for the authors to conduct additional experimental trials. An acid hydrolysis procedure, previously reported, could be carried out. https://doi.org/10.1016/j.aca.2018.09.037

Additional comments:

In lines 75-77, the authors should indicate the total number of this study cohort, showing the total numbers of PTM and EPTB patients.

In the section titled “Evaluation of clinical urine specimens using immunoassays,” the authors should mention the epitope that both antibodies FIND28 and S4-20 recognize in LAM. Also in this section, the authors mentioned that the LAM used for the standard curve was derived from the in vitro culture of the TB strain Aoyama B. The authors should explain or shows evidence to select this strain considering changes in LAM structure and specificity of antibodies compared with other strains.

In lines 50-51, the statistical of people with TB disease should be updated, considering the latest WHO report (2023).

Figures and Tables:

Figures 2 and supplementary figure 1: Information associated with the colors in the graph needs to be included. There is no legend describing the image observed.

Reviewer #2: This is a well written and methodologically rigorous paper which addresses an important clinical question and reports useful negative findings. I suggest a few minor revisions to improve the clarity and readability of this paper.

Firstly, the use of terminology throughout the paper needs to be addressed. Terminology on disease states should be brought in line with the TB-ICE consensus classification (https://doi.org/10.1016/S2213-2600(24)00028-6) and the reference to wards in PR-PTBW and PR-EPTBW is unclear and should be clarified.

Secondly, it is unclear what threshold/cut-off has been used to calculate performance metrics throughout and why this cut-off was used. AUROC is a better performance metric as takes into account different thresholds. The optimal/minimal WHO Target Product Profiles are not mentioned in the manuscript but must be as a benchmark.

Finally the link to the publicly accessible data on Dataverse is not working.

Below are some further detailed suggestions:

Line 29 - Switch from "TB-related symptoms" to suspected clinical TB throughout

Line 41 - Make clear that these are AUROC values

Line 64 - Sentence starting 'As such, the only...' does not follow from the previous sentence and should be amended/deleted

Line 71 - Suggest elaborating on the mixed results in reference 21

Line 86 - The TB ward terminology is confusing, especially considering some were recruited from outpatients. Suggest switching throughout to suspected PTB and suspected EPTB.

Line 164 - For the CRS did you address patients lost-to-follow-up? Were there any patients LTFU? If not, should be clarified in the results.

Line 171 - Switch from "presented for additional information" to "secondary analyses".

Line 176 - Unclear when unpaired vs paired ROC comparisons were performed. Paired comparison should be used.

Line 180 - The link to the publicly accessible code is not working, hence unable to verify if all data is available

Line 207 - How did you address those with both pulmonary and extra-pulmonary TB - ie the 7.5% of EPTB pts with a positive CXR? How were they classified?

Line 293 - The abbreviations in this sentence are not clear.

Line 223 - In table 1, switch from "History of recent TB exposure" to "Household contact". Additionally "Culture positive" and "CRS" should be separate rows.

Line 224 - It is unclear what being compared in this sentence.

Line 253 - It is not clear what theshold/cut-off has been used to calculate the accuracy throughout the results. This should be clarified with reasoning as to why it was chosen.

Line 260 - Sentence not clear, review.

Line 274 - Table 3 is very large and may be better in the supplementary materials, with a smaller table containing salient results (N, sensitivity, specificity, AUROC) in the main manuscript. Alternatively the entire table can be moved to supplementary and Table 4 expanded to include CRS.

Line 286 - Some subheadings would be useful in the results section.

Line 290 - I do not think "appeared to be slightly higher" is appropriate with n=6. Suggest rephrasing this sentence to something like "a comparison of Ab pairs was limited in the EPTB MRS positive group which contained only 6 individuals (50% vs 33%, p>0.99)".

Line 296 - In Figure 3 I suggest that this is split into two side-by-side plots for PTB and EPTB, and 95% CI are added.

Line 317 - Pivot from "presumptive TB disease" to "presumed clinical TB" in line with ICE-TB consensus classification (https://doi.org/10.1016/S2213-2600(24)00028-6).

Line 319- The AUROC is more illustrative of the performance of the test across different thresholds and should be mentioned before sensitivity/specificity here.

Line 322 - Subclinical-TB is a reference to absence of signs/symptoms as per ICE-TB consensus classification so suggest changing to paucibacillary disease.

Line 421 - This is the first time the Target Product Profiles are mentioned. This needs to be included in the introduction, with abbreviation defined.

7. PLOS authors have the option to publish the peer review history of their article (what does this mean?). If published, this will include your full peer review and any attached files.

**Do you want your identity to be public for this peer review?** For information about this choice, including consent withdrawal, please see our Privacy Policy.

Reviewer #1: **Yes: **Leslie Chavez-Galan, Ph.D.

Reviewer #2: **Yes: **James Greenan-Barrett

---

## [Decision Letter · Decision Letter 2]

13 Aug 2024

PGPH-D-23-02171R2

Urine lipoarabinomannan concentrations among HIV-negative adults with pulmonary or extrapulmonary tuberculosis disease in Vietnam

Dear Dr. Boyle,

Thank you for submitting your manuscript to PLOS Global Public Health and for addressing the reviewers' comments. The manuscript is very nearly suitable for acceptance, but there are some small residual issues itemised below. We invite you to submit a revised version of the manuscript that addresses the points raised:

1. As highlighted by the Reviewer, there is some residual use of the "wards" terminology and abbreviations, including in the Abstract. 

2. The new sample size statement (line 196) is ambiguous and requires clarification - what diagnostic accuracy metrics is this based on?

3. In Figure 2 (the density plots), these may benefit from being stratified by the primary MRS status

We look forward to receiving your revised manuscript.

Kind regards,

Rishi Kumar Gupta

Academic Editor

Journal Requirements:

Additional Editor Comments (if provided):

Reviewers' comments:

Reviewer's Responses to Questions

**Comments to the Author**

1. If the authors have adequately addressed your comments raised in a previous round of review and you feel that this manuscript is now acceptable for publication, you may indicate that here to bypass the “Comments to the Author” section, enter your conflict of interest statement in the “Confidential to Editor” section, and submit your "Accept" recommendation.

Reviewer #2: All comments have been addressed

2. Does this manuscript meet PLOS Global Public Health’s publication criteria? Is the manuscript technically sound, and do the data support the conclusions? The manuscript must describe methodologically and ethically rigorous research with conclusions that are appropriately drawn based on the data presented.

Reviewer #2: Yes

3. Has the statistical analysis been performed appropriately and rigorously?

Reviewer #2: Yes

4. Have the authors made all data underlying the findings in their manuscript fully available (please refer to the Data Availability Statement at the start of the manuscript PDF file)?

Reviewer #2: Yes

5. Is the manuscript presented in an intelligible fashion and written in standard English?

Reviewer #2: Yes

6. Review Comments to the Author

Reviewer #2: It was a pleasure to read the authors' response to the reviewer comments. The changes that they have made have significantly improved the readability of the manuscript. The terminology is clear throughout, there is a clear explanation for the threshold values used to determine positive tests, AUROC values have been compared and there is a clear explanation for when/why paired and unpaired ROC tests were performed. The dataverse link is now functioning and the data is publicly accessible.

I have two very minor typographical suggestions. Firstly there is some residual use of previous abbreviation PR-PTBW on line 116, 189, 294. Secondly on line 252 there is a bracket missing after "(n=230".

Otherwise, I believe this manuscript should be accepted for publication.

7. PLOS authors have the option to publish the peer review history of their article (what does this mean?). If published, this will include your full peer review and any attached files.

**Do you want your identity to be public for this peer review?** For information about this choice, including consent withdrawal, please see our Privacy Policy.

Reviewer #2: **Yes: **James Greenan-Barrett

---

## [Editor Report · Decision Letter 3]

20 Aug 2024

PGPH-D-23-02171R3

Urine lipoarabinomannan concentrations among HIV-negative adults with pulmonary or extrapulmonary tuberculosis disease in Vietnam

Dear Dr. Boyle,

Thank you for submitting your manuscript to PLOS Global Public Health.

Apologies if our previous feedback was unclear but we recommend that Figure 2 is presented with colours according to MRS status, and separate panels for the different assays and PTB vs EPTB. This would enable the distributions to be compared by MRS status within each panel. There should be 4 panels in total: PTB F/A, EPTB F/A, EPTB S/A, EPTB S/A. We hope this is now clear.

In addition, the abstract submitted within the editorial manager system does not appear to have been updated. We are happy with the other amendments. 

We look forward to receiving your revised manuscript.

Kind regards,

Rishi Kumar Gupta

Academic Editor
---

## [Editor Report · Decision Letter 4]

11 Oct 2024

Urine lipoarabinomannan concentrations among HIV-negative adults with pulmonary or extrapulmonary tuberculosis disease in Vietnam

PGPH-D-23-02171R4

Dear Dr Boyle,

We are pleased to inform you that your manuscript 'Urine lipoarabinomannan concentrations among HIV-negative adults with pulmonary or extrapulmonary tuberculosis disease in Vietnam' has been provisionally accepted for publication in PLOS Global Public Health.

Best regards,

Rishi Kumar Gupta

Academic Editor